# Mixed Primary Hepatocellular Carcinoma and Hepatic Neuroendocrine Carcinoma: Case Report and Literature Review

**DOI:** 10.3390/medicina59020418

**Published:** 2023-02-20

**Authors:** Woo Young Shin, Keon Young Lee, Kyeong Deok Kim

**Affiliations:** Department of Surgery, Inha University College of Medicine, Incheon 400-711, Republic of Korea

**Keywords:** hepatocellular carcinoma, neuroendocrine carcinoma, surgical resection

## Abstract

Mixed hepatocellular carcinoma with neuroendocrine carcinoma (HCC-NEC) is extremely rare, comprising about 0.46% of primary hepatic tumors. A 63-year-old man who was a chronic alcoholic presented with a nine-centimeter-sized hepatic mass. His serum alpha-fetoprotein and protein induced by vitamin K antagonist-II levels were 22,815 ng/mL and 183 mAU/mL, respectively. The patient underwent a right hemihepatectomy, including the middle hepatic vein. The tumor consisted of poorly differentiated HCC (20%) and large- and small-cell-type NEC (80%) components as per the pathological examination. Immunohistochemically chromogranin and synaptophysin were positive in the areas of NEC and negative in the areas of HCC. Adjuvant chemotherapy with a combination of cisplatin and etoposide was administered after surgery. At postoperative 5 months, the patient complained of right flank pain, and CT showed a new mass measuring 7.3 cm in the right adrenal gland. Postoperatively, after 6.5 months, more recurred masses were noted on the posterior aspect of the right kidney and both lungs. Although the regimen was changed from etoposide to irinotecan, additional recurred masses were developed in the liver, lung, and brain. He passed away 12 months after the surgery. After reviewing and analyzing previous literature, the 1 and 2 year overall survival rates are 57.3 and 43.6%, respectively, and the 1 and 2 year disease-free survival rates are 36.2 and 29.0%, respectively. Mixed HCC-NEC is a very rare tumor, and the surgical outcome is poor.

## 1. Introduction

Hepatocellular carcinoma (HCC) is the most common primary liver cancer. Sometimes, HCC can combine with other cell types, of which cholangiocarcinoma is the most common one [1]. In contrast, HCC with neuroendocrine carcinoma (HCC-NEC) is very rare, comprising about 0.46% of primary hepatic tumors [2]. Twenty-nine cases of primary HCC-NEC have been reported in the literature until now. The prognosis seemed to be poor. However, some authors have reported a favorable prognosis. Here, we reported a patient with HCC-NEC with a poor prognosis. Additionally, we analyzed previous reports, including our case.

## 2. Case Presentation

A 63-year-old man presented with a hepatic mass that had been detected by computed tomography (CT) at another clinic. He drank about 36 g of alcohol per week for 40 years. The routine laboratory test and liver function tests showed normal values. The viral marker showed negative serum hepatitis B surface antigen (HBsAg) and hepatitis C antibody (anti-HCV), and positive serum hepatitis B antibody (HBsAb). His serum alpha-fetoprotein (AFP) level was very high, at 22,815 ng/mL, and his protein induced by vitamin K antagonist-II (PIVKA-II) level was elevated to 183 mAU/mL (normal range, <40 mAu/mL). Other tumor markers such as carcinoembryonic antigen (CEA) and carbohydrate antigen19-9 (CA19-9) levels were also slightly increased to 6.1 ng/mL and 80.9 U/mL, respectively. The prior CT identified a nine-centimeter-sized mass in the anterior section. Additional liver ethoxybenzyl diethylenetriamine pentaacetic acid-enhanced magnetic resonance imaging revealed heterogeneous enhancement in the arterial phase and delayed wash-out in the delayed phase (Figure 1). Another systemic metastasis was not found. Therefore, the patient underwent a right hemihepatectomy, including the middle hepatic vein. During the pathological examination, the cut surface showed a multinodular solid mass, measuring 9.0 × 8.0 × 6.0 cm. The mass was a well-demarcated, yellow-whitish, soft, and firm one with multifocal necrosis and hemorrhage (Figure 2). The tumor consisted of poorly differentiated HCC (20%) and large- and small-cell-type NEC (80%) components. The tumor invaded the right anterior portal vein and microvessels. There was no lymph node metastasis. Immunohistochemically chromogranin and synaptophysin were positive in the areas of NEC and negative in the areas of HCC (Figure 3). In contrast, AFP and hepatocyte antigen were positive in the areas of NEC and HCC. Adjuvant chemotherapy with a combination of cisplatin and etoposide was administered after the surgery. At 5 months post-operation, the patient complained of right flank pain, and CT showed a new mass measuring 7.3 cm in the right adrenal gland. The patient received radiotherapy (5000 cGy) for the recurred mass. After 6.5 months post-operation, the adrenal mass disappeared, but other recurred masses were noted on the posterior aspect of the right kidney and both lungs. Although the regimen was changed from etoposide to irinotecan, additional recurred masses developed in the liver, lung, and brain. He passed away 12 months after the surgery.

## 3. Discussion

Primary hepatic NEC is very rare, with an incidence of 0.46% among primary hepatic malignancies [1]. On the other hand, HCC is the most common hepatic malignancy. Sometimes, HCC is often combined with other malignancies. Combined HCC and cholangiocarcinoma is the most common type, accounting for 2.0–3.6% of primary hepatic malignancies [2]. Mixed HCC-NEC is extremely rare. Including our case, only 30 cases have been reported to date.

Two hypotheses have been proposed for the origin of primary hepatic NEC [3]. First, neuroendocrine cells in the intrahepatic bile duct epithelium undergo malignant transformation and become NEC cells. Second, NEC originates from hepatic stem cells that have dedifferentiated from other malignant hepatic cells and converted into neuroendocrine cells. In our case, the NEC cells were positive for immunostaining of AFP and hepatocyte antigen, which could support the latter hypothesis.

In Table 1, we have summarized the previously reported cases. Twenty-seven (90%) cases were of men, and the mean age is 62.7 (range: 19–84). None of the cases were diagnosed before the biopsy or surgery because of their rarity. Except for 2 non-operable cases, 28 cases underwent surgical resection. Of these, 15 (53.6%) cases recurred. Most of the patients recurred within one year regardless of the adjuvant therapy. Seven cases did not experience recurrence during the follow-up periods. Among these, the follow-up period for three cases was less than one year, so the follow-up period was too short to conclude that they did not recur. In the remaining four patients who did not recur during a follow-up period of more than 18 months, the tumor sizes were below 5 cm. However, given the early recurrence of other patients with small-sized tumors, other tumor factors such as vascular invasion are thought to affect recurrence. Our case had portal vein invasion and recurred in the adrenal gland, liver, lungs, and brain.

There has been no established treatment for HCC-NEC. According to previous reports, surgical resection seemed to be the most effective one. Additionally, an effective chemotherapy regimen has not been established. The longest survivor received adjuvant chemotherapy with a combination of cisplatin and etoposide after surgery [11]. So, four authors and ourselves used the same regiment, but all of the cases recurred [15,16,17,26]. After recurrence, thalidomide and bevacizumab, sorafenib, and nivolumab were used, but none of them prevented disease progression [9,21,26]. In our patient, radiotherapy did not prevent recurrence, but it was somewhat effective against the target tumor. According to the reports so far, it was difficult to treat the recurred patients. The diseases progressed until the end of the follow-up. The cumulative 1 year survival rate of 25 reported cases was 58% [3]. According to our evaluation, the 1 and 2 year disease-free survival rates were 36.2 and 29.0%, respectively (Figure 4).

In conclusion, mixed HCC-NECs are extremely rare tumors. The clinical courses are aggressive, and the prognosis is poor. Though, early surgery by early detection is thought to be the only way to expect long-term survival. Further, the accumulation of cases is required to establish effective treatment modalities.

## Figures and Tables

**Figure 1 medicina-59-00418-f001:**
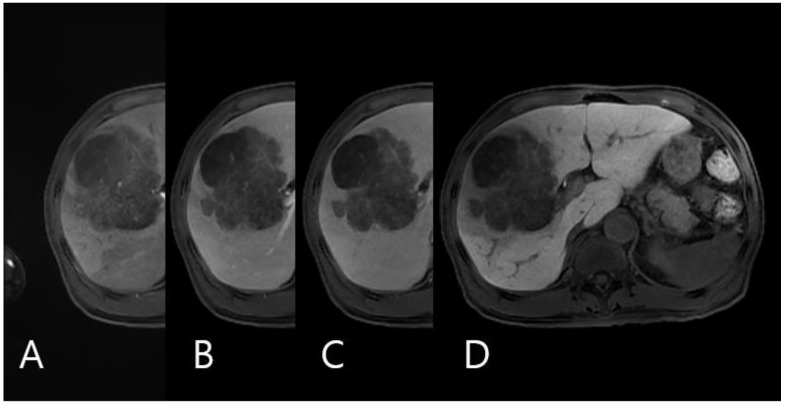
Liver ethoxybenzyl diethylenetriamine pentaacetic acid-enhanced magnetic resonance imaging Images. (**A**) Arterial phase, (**B**) portal phase, (**C**) venous phase, (**D**) and hepatobiliary phase.

**Figure 2 medicina-59-00418-f002:**
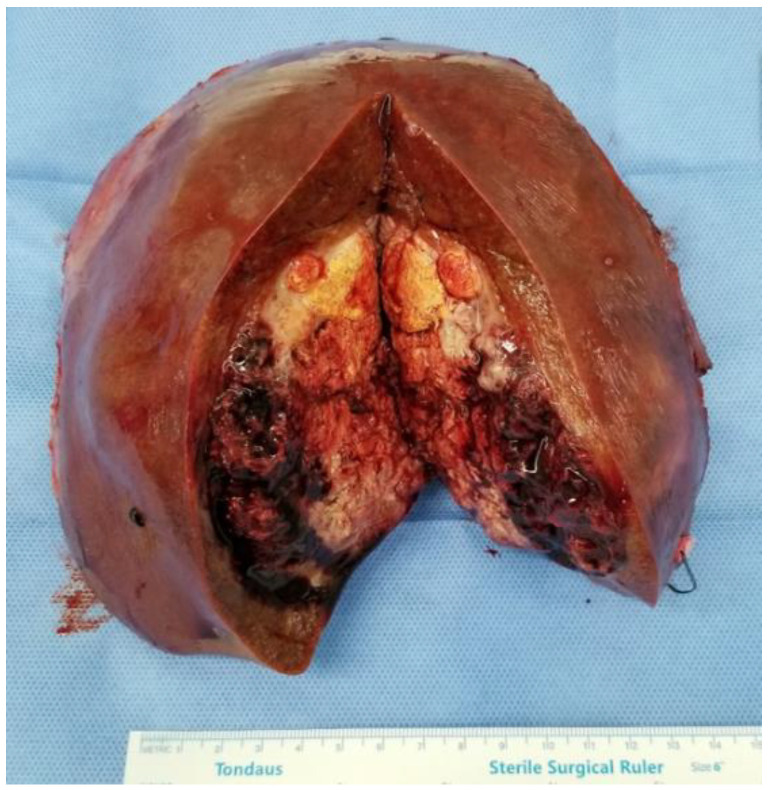
Gross specimen of the right hemihepatectomy. A nine-centimeter-sized tumor with multifocal necrosis and internal hemorrhage.

**Figure 3 medicina-59-00418-f003:**
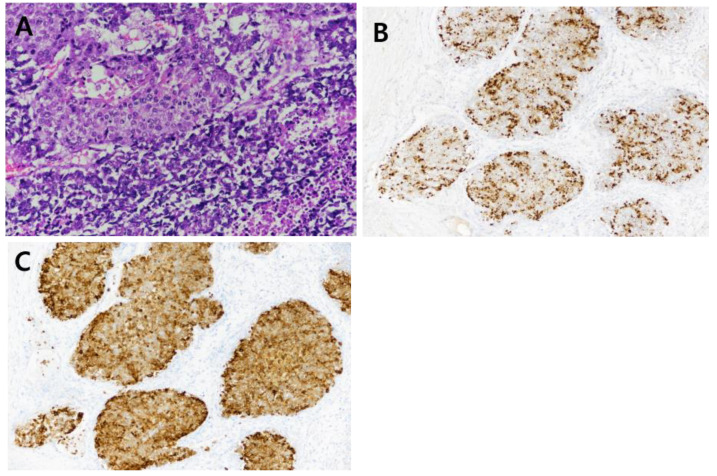
Microscopic findings. Hematoxylin and eosin stain demonstrated the intermingled HCC (top) and the large- and small-cell NEC (bottom) of the image ((**A**), magnification ×200). The NEC stains positive for chromogranin ((**B**), magnification ×100) and synaptophysin ((**C**), magnification ×100).

**Figure 4 medicina-59-00418-f004:**
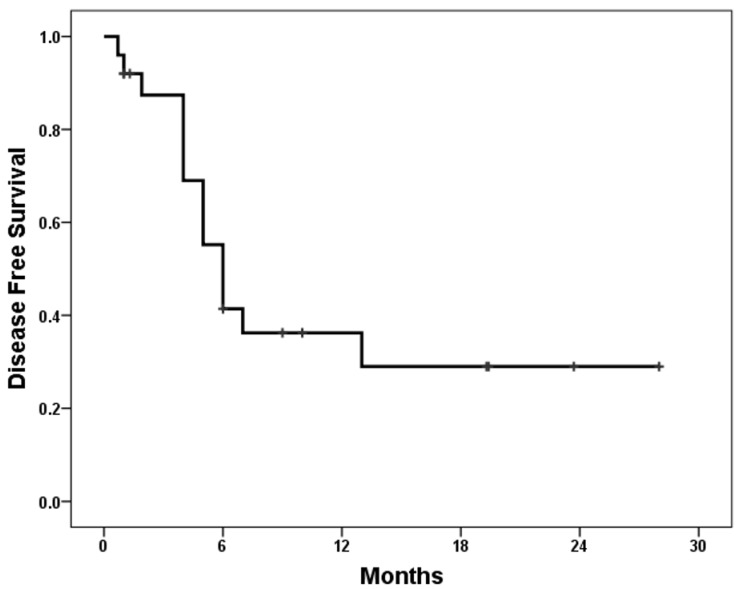
Disease-free survival of HCC-NEC after surgical resection; 1 and 2 year disease-free survival rates were 36.2 and 29.0%, respectively.

**Table 1 medicina-59-00418-t001:** Cases of mixed hepatocellular carcinoma and neuroendocrine carcinoma.

	Age/Sex	Tumor Size (cm)	Type	Initial Treatment	Recurrence	Survival
Barsky et al. [4]	43/M	Large	Combined	Palliative CTx d/t omental seeding		Dead (26 mon)
Artopoulos et al. [5]	69/M	10	Combined	Resection	NA	NA
Vora et al. [6]	63/M	10	Combined	Resection	NA	Dead (1 mon)
Ishida et al. [7]	72/M	3	Collision	Resection	NA	NA
Yamaguchi et al. [8]	71/M	4.1	Combined	Resection	Recurrence (bone, 5 mon)	Alive (F/U 5 mon)
Garcia et al. [9]	50/M	5.3	Collision	Resection	Recurrence (liver, peritoneum 4 mon)	Alive (F/U 16 mon)
Yang et al. [10]	65/M	7.5	Combined	Resection	Recurrence (liver and adrenal, LN 4 mon)	Dead (12 mon)
Tazi et al. [11]	68/M	4	Collision	Resection	No recurrence	Alive (F/U 28 mon)
Nakanishi et al. [12]	76/M	3	Combined	TACE followed by resection	Recurrence (bone, 7 mon)	Dead (17 mon)
Hammedi et al. [13]	51/M	20	Combined	Resection	NA	Dead (1 mon)
Aboelenen et al. [14]	56/F	7.5	Combined	Resection	No recurrence	Alive (F/U 6 mon)
Nishino et al. [15]	72/M	2.5	Combined	Resection	Recurrence (LN, 0.7 mon)	Dead (2 mon)
Yun et al. [16]	68/F	2.5	Combined	Resection	Recurrence (bone, 6 mon)	Dead (13 mon)
Choi et al. [17]	72/M	2.5	Collision	Resection	Recurrence (liver, 6 mon)	Alive (F/U 10 mon)
Beard et al. [18]	19/M	25	Combined	Resection	Recurrence (LN, 4 mon)	Alive (F/U 8 mon)
Nomura et al. [1]	71/M	4	Combined	Resection	Recurrence (Liver, NA)	Dead (10 mon)
	71/M	3	Collision	RFA followed by resection	Recurrence (Liver, NA)	Dead (8.6 mon)
	58/M	4.3	Combined	Resection	No recurrence	Alive (F/U 19.4 mon)
	50/M	1.8	Combined	Resection	No recurrence	Alive (F/U 19.3 mon)
	63/M	3	Combined	Resection	No recurrence	Alive (F/U 23.7 mon)
Baker et al. [19]	76/M	5.5	Collision	Resection	NA	NA
Liu et al. [20]	65/M	4.3	Collision	Resection	Hepatic and renal failure	Dead (1.3 mon)
Okumura et al. [21]	70/M	11	Mixed	Resection	Recurrence (bone, 1 mon)	Dead (3 mon)
Lu et al. [22]	65/M	14	Combined	Hospice		NA
Yilmaz et al. [23]	56/M	2.3	Collision	Resection	No recurrence	Alive (F/U 10 mon)
Ikeda et al. [24]	79/M	6	Combined	Resection	Recurrence (NA, 4 mon)	Dead (4 mon)
Kwon et al. [25]	44/M	10.5	Combined	Resection	Recurrence (liver and bone, 1.9 mon)	Dead (4 mon)
Jahan et al. [26]	50/M	2.7	Combined	Resection	Recurrence (liver, LN, and bone, 13 mon)	Dead (33 mon)
Nakano et al. [3]	84/F	5.5	Combined	Resection	No recurrence	Alive (F/U 9 mon)
Present case	63/M	9	Combined	Resection	Recurrence (adrenal, liver, and lung, 5 mon)	Dead (12 mon)

Abbreviations: CTx: chemotherapy, F: female, F/U: follow-up, LN: lymph node, M: male, NA: not applicable, RFA: radiofrequency ablation, TACE: transarterial chemoembolization.

## Data Availability

No new data were created or analyzed in this study. Data sharing is not applicable to this article.

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
