# Peer review of "Mixed Primary Hepatocellular Carcinoma and Hepatic Neuroendocrine Carcinoma: Case Report and Literature Review"

_medicina, 2023, doi:10.3390/medicina59020418_

Round 1

Reviewer 1 Report

Good case, relevant and interesting. I think it adds to the literature. 

Some questions: 

- What was the presenting symptom? Why did the patient have the CT scan at another clinic?

- Was there any genetic testing? What was the Ki-67 if measured?

- I like the table in the discussion but is there a chance to add the chemo regimen used in each study after the recurrence if available as well as any genetic findings if available?

Author Response

Point 1: What was the presenting symptom? Why did the patient have the CT scan at another clinic?

 Response 1: The patient had no symptoms. The mass was discovered incidentally during a medical check-up.

Point 2: Was there any genetic testing? What was the Ki-67 if measured?

Response 2:

Sorry, we did checked genetic test and Ki-67.

Point 3: I like the table in the discussion but is there a chance to add the chemo regimen used in each study after the recurrence if available as well as any genetic findings if available?

Response 3:

Four articles described chemo-regimen after relapse.

One author maintained etomidate and cisplatin. The others changed regimen - thalidomide and bevacizumab, sorafenib, and nivolumab.

I added this to the discussion.

Only Baker et al. performed genetic analysis.

So it would be better to describe the genetic factors after collecting more findings.

Reviewer 2 Report

I have two suggestions and three questions to the authors:

Suggestions:

1) the paper will benefit from moderate English editing and correction of typos.

2) Picture of Figure 2 is in the sport of Figure 3n and the histology of Figure 3 is in the spot of Figure 2. Please, change appropriately. 

Questions:

1) Was the recurrent tumors from NET, from HCC, or both? 

2) Since there was PV vascular invasion, was there a thought of immunotherapy?

3) How can we be sure this is a mixed tumor (HCC-NET) and not a concomitant NET in the background of HCC-alcoholic liver fibrosis?

Author Response

Suggestions:

  • the paper will benefit from moderate English editing and correction of types.

Response : Thanks

2) Picture of Figure 2 is in the spot of Figure 3 and the histology of Figure 3 is in the spot of Figure 2. Please, change appropriately. 

Response : I fixed them

Point 1: Was the recurrent tumors from NET, from HCC, or both? 

Response 1:

Preoperative AFP and PIVKA II levels highly increased, but thye not increase after early recurrence. So, I thought that the recurrent tumors were from NEC.

 Point 2: Since there was PV vascular invasion, was there a thought of immunotherapy?

Response 2:

 No, we did not consider the immunotherapy.

Point 3: How can we be sure this is a mixed tumor (HCC-NET) and not a concomitant NET in the background of HCC-alcoholic liver fibrosis?

Response 3:

Well, it is a difficult question.

If it’s a matter of the histogenesis of the tumor, it seems difficult to answer because of insufficient and inconsistent genetic information.

But, Okumura et al. reported that the NEC component of the combined type could be formed by transdifferentiation of the HCC component, and the NEC in the collision-type tumor part could be speculated as metastasis of fully transformed NEC of the combined-type tumor part.
